# Comparative Antioxidant Activity and Untargeted Metabolomic Analyses of Sour Cherry Cultivars Based on Ultra-Performance–Time of Flight–Mass Spectrometry

**DOI:** 10.3390/plants13111511

**Published:** 2024-05-30

**Authors:** Prabhjot Kaur, Ahmed G. Darwish, Islam El-Sharkawy, Ashutosh Singh, Jayasankar Subramanian

**Affiliations:** 1School of Engineering, University of Guelph, Guelph, ON N1G 2W1, Canada; pkaur07@uoguelph.ca (P.K.); asingh47@uoguelph.ca (A.S.); 2Center for Viticulture and Small Fruit Research, College of Agriculture and Food Sciences, Florida A&M University, Tallahassee, FL 32308, USA; ahmed.darwish@famu.edu (A.G.D.); islam.elsharkawy@famu.edu (I.E.-S.); 3Department of Biochemistry, Faculty of Agriculture, Minia University, Minia 61519, Egypt; 4Department of Plant Agriculture, University of Guelph, Guelph, ON N1G 2W1, Canada

**Keywords:** sour cherry, metabolomics, antioxidant activity, phenolics, functional foods

## Abstract

This study was conducted for the comparative analysis of antioxidant activity and untargeted metabolomics of dark- and light-colored sour cherry cultivars grown in Canada. Based on our previous study, we selected four cultivars—‘Heimann R’, ‘Gorsemska’, V70142, and ‘Montmorency’—to determine the untargeted metabolites and their role in antioxidant activities. A total of 473 metabolites were identified from four sour cherry genotypes using UPLC–ToF–MS. Untargeted metabolomics revealed the dominant chemical groups present in sour cherries. PCA showed that the diversity in sour cherry metabolites was due to the genotype differences indicating iditol, malic acid, chlorobenzene, 2-mercaptobenzothiazole, and pyroglutamic acid as the predominant contributors. The variable importance in the projection (VIP > 1.0) in partial least-squares–discriminant analysis described 20 biomarker metabolites representing the cherry metabolome profiles. A heatmap of Pearson’s correlation analysis between the 20 biomarker metabolites and antioxidant activities identified seven antioxidant determinants that displayed the highest correlations with different types of antioxidant activities. TPC and TAC were evaluated using the Folin–Ciocalteu method. The total antioxidant activity was performed using three different assays (ABTS, FRAP, and DPPH). This study of correlating metabolomics and antioxidant activities elucidated that the higher nutritional value and biological functions of sour cherry genotypes can be useful for the development of nutraceutical and functional foods.

## 1. Introduction

Cherry is a member of the Rosaceae family, belonging to the genus *Prunus*. Although several *Prunus* species are identified as cherry, two species—*Prunus avium* L. (sweet cherry) and *Prunus cerasus* L. (sour cherry)—are the most common edible species [1]. Sour cherries are mostly processed into value-added products such as jam, juices, cannery products, and alcoholic beverages while sweet cherries are consumed as fresh. Sour cherries are a rich source of polyphenolic compounds associated with antioxidant activity and other health benefits [2]. Fruit phenolic compounds include anthocyanins, the primary determinant of color, and the secondary metabolites including phenolic components (flavonols, flavonoids, flavones, flavanones, catechin, and epicatechin), known for their therapeutic properties [3] and for mitigating certain cancers [4,5]. The consumption of sour cherries helps in treating several health issues such as muscle degeneration [6], heart ailments [7], cancer, diabetes [8], and inflammation [9]. In addition, its juice or concentrate consumption reduces the risk of gout attacks and arthritis, making it an attractive functional food [10].

According to FAO 2020, the sour cherry production in North America was 66,425 tonnes, with 4236 tonnes of production in Canada [11]. Many sour cherry cultivars cultivated in North America are of European origin [12,13]. In North America, ‘Montmorency’ is the most common sour cherry cultivar grown commercially [14]. Hence, existing sour cherry breeding programs focus on increasing fruit quality traits such as taste, appearance, nutraceutical value, and production yield. Nowadays, due to increased interest in the emerging health benefits of fruits, agricultural breeding programs focus more on secondary metabolites [15]. Previously, we analyzed the physicochemical characteristics of a diverse group of sour cherry genotypes grown in Canada [16]. From that work, we selected four sour cherry cultivars—two dark-flesh and two light-flesh cultivars—including ‘Montmorency’, which is considered the standard because it is the only genotype in North America for metabolic profiling.

Fruits of sour cherries contain major chemical groups, including sugars such as glucose, organic acids like citric and malic acid, and polyphenolic compounds (phenolic acids, flavonols, flavonoids, and anthocyanins), which could be influenced by genotypes and the environment [17]. Although it is generally agreed that sour cherries are rich sources of metabolites, for some reason, they have not received the attention of other fruit crops of the same class, for instance, as blueberries. Sour cherry juice provides several health benefits, such as anti-inflammatory and anti-gout, and has been claimed to be effective in decreasing CRP (C-reactive protein) and oxidative stress [17]. Of late, sour cherry juice consumption is also gaining public attention as a superfood or nutraceutical based on people’s attestations.

While sour cherries are rich in metabolites, to date, no comprehensive study analyzing the complete metabolic profile of sour cherries grown in North America has been reported. Hence, we aimed to explore the metabolic profile of four sour cherry cultivars with diverse genetic background and their cross-correlation with antioxidant activity. We used ultra-performance–time of flight–mass spectrometry (UPLC–ToF–MS) to perform untargeted metabolomics to analyze the sour cherry samples. UPLC–ToF–MS is a powerful tool with a shorter retention time [18]. We chose this technique as it is convenient for the detection of similar metabolites of lower concentration [19]. Further, the quantitative analysis of bioactive components (total phenols and antioxidants) was performed to correlate it with metabolic components. To our knowledge, this is the first detailed report of the metabolomic profile of sour cherry cultivars grown in North America.

## 2. Results and Discussion

Non-targeted metabolic analysis with UPLC–ToF–MS provides a better and accurate quantitative analysis of plant metabolites responsible for the targeted profiling of total phenolics and antioxidants [20]. In this study, analysis using UPLC–ToF–MS was performed to separate and characterize the major bioactive components in sour cherry cultivars grown in North America. The flesh and juice color of these cultivars are shown in Figure 1A and Figure 1B respectively. Further, PCA and PLS statistical analyses were used to determine the correlation between these metabolites and antioxidant activity.

### 2.1. Quantification of Total Phenolic, Anthocyanin, and Antioxidant Content

The total phenolic content (TPC) of the dark-flesh cultivars ranged from 2.79 to 2.82 mg GAE/g FW (gallic acid equivalent), while light-flesh genotypes had a range of 1.46–1.98 mg GAE/g FW. Overall, the findings suggest that there is a meaningful and statistically significant difference (*p* < 0.05) in TPC between the two categories of sour cherry genotypes based on flesh color (Figure 2A). The red color of the cherry flesh is due to the presence of a major anthocyanin component: cyanidin-3-O-rutinoside [1]. TPC and total anthocyanin content (TAC) values also followed a similar pattern for dark- and light-flesh cultivars. The average TAC value for the dark-flesh cultivars ranged between 3.05 and 4.85 mg, and between 0.38 and 0.94 mg cyanidin-3-glucoside Eq/g FW in the light flesh cultivars. TPC in dark flesh cultivars similar to fruits like blueberry, blackberry, and haskap was higher because of the accumulation of a high amount of phenolic acids—hydroxycinnamic acids, flavonols, and anthocyanin components [1].

The antioxidant activity of sour cherry extract is strongly related to the structural activity of the metabolite composition as presented in Table 1. Thus, the antioxidant potential of the extract measured using different assays would provide various aspects of scavenging activity [15,18]. Antioxidant activity assessed by FRAP, ABTS, and DPPH followed the same pattern as observed for total phenolics, flavonoids, and anthocyanin components (Figure 2B).

Dark-flesh cultivars showed two times more antioxidant activity than light-flesh cultivars. The antioxidant content for the dark-flesh cultivars is approximately two-fold higher than the previously reported results [19] as well as possibly due to higher anthocyanin and total phenolic components [20].

Further, the radical scavenging activity assessed by DPPH indicates that the antioxidant potential of dark-colored cultivars is lower compared to light-colored cultivars. These results are in correspondence with the Hungarian dark-colored cultivars [21] and the muscadine grape genotype [22]. This is because of the negative potential of degrading the purple color of the DPPH reagent to yellow color.

### 2.2. Untargeted Metabolic Profiling of Selected Sour Cherry Cultivars

Metabolites are the small molecules obtained from the different biological cells, tissues, and biofluids with metabolic processes and can be used as a chemical marker. These metabolite markers can be identified by targeted and untargeted metabolomics. So, in this study, we tried to perform untargeted metabolomic extraction. A total of 473 metabolites (279 in ESI+ mode, 155 in ESI− mode, and 39 in both ESI+ and ESI− mode) were identified in sour cherry cultivars (Appendix A). Metabolite set enrichment analysis (MSEA) was performed to identify the primary chemical groups associated with all the 473 identified metabolites representing the significant chemical group’s classification of identified metabolite sets (top 25) with enrichment ratio (Figure 3A,B). The important groups identified in metabolites were organic acids, organoheterocyclic compounds, benzenoids, carbohydrates, nucleic acids, polyketides, organic oxygen, and nitrogen compounds, and a broad suite of lipid molecules such as fatty acids, prenol lipids, sterol lipids, sphingolipids, glycerolipids, lipids and lipid-like molecules, and glycerophospholipids (Figure 3B).

Previous studies indicated that the primary chemical components found in sour cherry were fruit sugars (fructose, glucose, and sucrose), organic acids (malic acid), phenolic acids (5-caffeoylquinic, 3-caffeoylquinic, and p-coumaric acids), flavonols (catechin and derivatives, quercetin, and kaempferol glycosides), and coloring pigments (anthocyanins) [17,21].

To obtain the possible differences in the bioactive components between dark- and light-flesh cultivars, we performed a multivariate statistical analysis. PCA 2D score plot (Figure 4A) showed a variation of 68.7% and 25.7% in the first (PC1) and second (PC2) principal components, respectively. A clear separation of all the cultivars based on different associated metabolites was observed. All the cultivars deviated from each other and clustered in different groups depending on metabolite similarities. As expected, a close aggregation was observed between dark-flesh cultivars (‘Heimann R’ and ‘Gorsesmka’) and light-flesh cultivars (‘Montmorency’ and V70142) which also showed the segregation from the dark-flesh cultivars.

Further, the PCA scatter plot analysis suggested that iditol was the primary metabolite contributing to the ‘Heimann R’, and malic acid was the predominant contributor to the ‘Gorsemska’. However, chlorobenzene, 2-mercaptobenzothiazole, and pyroglutamic acid were the dominant donors for the standard variety (‘Montmorency’) (Figure 4B). Despite identifying all the metabolites in the four cultivars, PCA analysis revealed that the difference in sour cherry metabolites was majorly due to the genotype differences. Our results for the metabolic information of dark-flesh cultivars ‘Heimann R’ and ‘Gorsemska’ are in correspondence with the results observed for Turkish sour cherry cultivars [23].

### 2.3. Correlation of Total Metabolites and Bioactive Components

Partial least-square–discriminant analysis (PLS-DA) of sour cherry cultivars was performed among the identified metabolites (473) of dark- and light-flesh cultivars. This analysis helped to obtain the biomarker metabolites based on variable importance in projection (VIP) scores [24]. PLS-DA analysis of identified metabolites demonstrated a total of 20 chemical markers iditol, 4-oxo proline, 3-indole acrylic acid, pyroglutamic acid, glutamic acid, catechin, indole, malic acid, chlorogenic acid, gluconic acid, 2,4-hexadienedial, 2-oxoglutaric acid, citric acid, biochanin a, 3,4-dihydroxybenzaldehyde, hydroxybutyric acid, allysine, serine, kaempferol, and acetylacrylic acid observed for dark- and light-flesh sour cherry cultivars with a VIP score of ≥1.0 (Figure 5). Many metabolites identified in the different genotypes are present in both the dark-flesh and the light-flesh groups of cherries, although they differ in their concentration. Notable changes in metabolites to the different flesh colors were observed, which involved malic acid and 3,4, dihydroxy benzaldehyde. These chemical metabolites were predominantly organic acids, phenolic acids, flavonoids, anthocyanins, and amino acids. Previous reports suggested that sour cherry phenolic components mainly contain phenolic acid derivatives (hydroxycinnamic acid, gallic acid, ascorbic acid, citric acid, chlorogenic acid, and neochlorogenic acid), flavonols (catechin, epicatechin, rutin, and quercetin), and anthocyanin components (cyanidin glucoside derivatives, kaempferol, and acetylacrylic) [25].

An untargeted metabolomics strategy can be used to identify the metabolome associated with the antioxidant activity present in sour cherry varieties having different flesh colors. To test this hypothesis, we performed a heatmap analysis of the major compounds identified in sour cherries (Figure 4). The heatmap results indicated a better visualization of the dark- and light-flesh cultivars’ metabolite content and antioxidant activity. Based on the FRAP and ABTS assay data presented in Table 1, the dark-flesh cultivars have a higher antioxidant capacity. DPPH assay suggested that one of the dark-colored cultivars—’Gorsemska’—is comparatively low in antioxidant capacity among the four cultivars [16]. This follows the data presented in Figure 1B as well, which show that the DPPH value, as determined by the standard colorimetric process, is indeed the lowest in ‘Gorsemska’. It has been observed earlier that sweet cherries that had high malic acid also exhibited high DPPH values [26]. This could be because the proton transfer reaction of DPPH^+^ free radical by a scavenger (A–H)—often influenced by organic acids—causes a decrease in absorbance and thus lowers the absorbance for DPPH assay [27]. Furthermore, Huang et al. [28] showed that malic acid applications decreased the DPPH activity in lychee fruits. Together, these results suggest that the change in malic acid content could be a factor in the reduced DPPH value in ‘Gorsemska’. A more recent study in Chinese cherry [*Prunus pseudocerasus* (Lindl.)] suggested that bitterness in these cherries is a result of mainly limocitrin-7-O-glucoside, with minor effects from eight other compounds [29].

The heatmap generated from the metabolomic analyses showed that the dark-flesh cultivars exhibited relatively higher TPC, and TAC values. The metabolites responsible for the high antioxidant activity in dark-flesh cultivars were biochanin A, allysine, 2,4-hexadienedial, and acetyl acrylic acid, while glutamic acid, pyroglutamic acid, and serine seem to control the antioxidant levels in light-flesh cultivars (Figure 5). Previous studies also reported that the major flavonoid and phenolic components present in sour cherry include chlorogenic acid, neochlorogenic acid, catechin, epicatechin, and epigallocatechin [30].

To further elaborate on the metabolites, a heatmap Pearson’s correlation analysis was also performed to identify the link between metabolites and antioxidant activity (Figure 6). The red–blue color in the plot corresponds to the higher–lower correlation between the components. The Pearson correlation coefficient cluster analysis also confirmed the antioxidant activity observed earlier by us for total phenolics, anthocyanins, and flavonoids [16].

An elaborate view of Pearson’s correlation coefficient among candidate metabolites with antioxidant activity and phenolic components is presented in Figure 7. The results notably revealed seven metabolites with higher ABTS and FRAP antioxidant possibility, including allysine, biochanin A, hydroxybutyric acid, acetyl acrylic acid, 2,4-hexadienedial, malic acid, and citric acid. In contrast, indole and chlorogenic acid were highly correlated with DPPH antioxidant activity. Several studies have reported the antioxidant potential of phenolic acids and their derivatives in fruits and vegetables [31]. Antioxidant analyses suggested that the activity occurred in a genotype-specific manner, despite fruit color. Among the evaluated assays, antioxidant activity results for the dark-flesh cultivars are approximately two-fold higher than the light-flesh cultivars, potentially due to higher levels of anthocyanin and total phenolic components [20,32]. Total phenolic content, total anthocyanins, and total flavonoids exhibited a high correlation (at *p* < 0.05) with FRAP and ABTS, which is consistent with our earlier report [16]. The multifaceted nature of these mechanisms such as ROS scavenging, chelating metal ions, inducing antioxidant enzymes, etc., underscores the versatility of phenolic acids in combating oxidative stress and highlights their potential health benefits [33]. It is important to note that the specific effects may vary depending on the type of phenolic acid and the biological context in which they are studied. The antioxidant potential for allysine is due to the actions of lysyl oxidase in the extracellular matrix, which also plays a key role in the crosslink formation to stabilize collagen and elastin [34]. Biochanin A, an isoflavone present in many plants (e.g., chickpea, red clover, and soybean), is increasingly sought after and potentially could lead to the development of nutraceuticals and pharmaceuticals, largely due to its antioxidant activities [35]. The antioxidant effect of hydroxybutyric acid is achieved by diminishing pro-oxidant oxidative stress markers like reactive oxygen species and enhancing glutathione content, resulting in decreased lipid peroxidation [36]; moreover, acetyl acrylic acid was reported to have antioxidant activities [37]. Malic acid and citric acid are widely distributed in fruit vinegar and exert health benefits such as antioxidant and antimicrobial activities, managing blood glucose, and regulating lipid issues [38]. Contrastingly, indole acids revealed a promising antioxidant activity [39]. Chlorogenic acid exhibits many biological properties, including antioxidant and antibacterial activities [40]. Our results suggested that the differences in antioxidant activity between the different cherry cultivars correlated with the accumulation of specific chemical compounds that differentiate between the dark/light-flesh cherry genotypes.

## 3. Materials and Methods

### 3.1. Chemicals and Reagents

Folin–Ciocalteu phenol reagent, 2,2′-azinobis-(3-ethylbenzothiazoline-6-sulfonic acid) (ABTS), gallic acid, trolox, cyanidine-3-glucoside, quercetin, 2,4,6-tripyridyl-s-triazine (TPTZ), catechin, epicatechin, rutin, ascorbic acid, chlorogenic acid, neochlorogenic acid, ferulic acid, and caffeic acid were purchased from Sigma (Sigma-Aldrich, Oakville, ON, Canada). HPLC-grade solvents—acetonitrile, methanol, acetic acid, and formic acid—were obtained from Fisher Scientific (Whitby, ON, Canada).

### 3.2. Cherry Samples

Fully ripe sour cherry fruits were hand-picked from the University of Guelph’s breeding program at Vineland Research and Innovation Centre, Vineland, Ontario in July 2021. Based on previous analysis, four cultivars, including two dark-flesh (‘Heimann R’ and ‘Gorsemska’) and two light-flesh (V70142 and ‘Montmorency’) (Figure 1A) ones, were used in this study. These varieties were selected based on their physicochemical properties and antioxidant capacities that we reported earlier [16]. After harvest, the fruits were stored for analysis at −80 °C for 6 months.

### 3.3. Preparation of Cherry Extract

Frozen cherry samples were powdered using a cryogenic grinder (Geno grinder, Metuchen, NJ, USA). Approximately 15 g of powdered sample was subjected to solvent extraction in 100 mL methanol, and the mixture was kept for shaking (200 rpm) at room temperature for 24 h in the dark. Then, the extracts were filtered through 0.45 µm cellulose fiber P8-grade filter paper (Fisher Scientific, Mississauga, ON, Canada). The obtained filtrate was concentrated using a rotary evaporator (IKA HB10, Cole-Parmar, Mississauga, ON, Canada) at 40 °C with an extraction yield of 10 g/100 g FW of fruit. Further, the residual solvent in extracts was evaporated in a fume hood to obtain the concentrated extract. The obtained extracts were stored at 4 °C for further analysis.

### 3.4. Determination of Total Phenolic, Total Anthocyanin, and Total Antioxidant Activity

Total phenolic content (TPC) was estimated as described earlier [18] using Folin–Ciocalteu reagent. Briefly, 50 µL of the diluted sour cherry extract was mixed with 50 µL of the 10% FC phenol reagent diluted in water; following 6 min of mixing, 100 µL of 7% Na_2_CO_3_ was added to the mixture and kept in the dark for 90 min. The absorbance of the sample was determined at 765 nm using a plate reader spectrophotometer (Biotek Synergy H1 Hybrid reader, Agilent Technology, Santa Clara, CA, USA) and measured as gallic acid equivalent. Total anthocyanin content (TAC) was estimated using the pH-differential spectroscopic method [10]. The total antioxidant activity was conducted using three different assays (ABTS, FRAP, and DPPH) [41]. Antioxidant capacity was assessed with DPPH, FRAP, and ABTS as per the previously published method [42]. The details of the extraction procedures have been described in an earlier report [16].

### 3.5. Sample Preparation for UPLC–ToF–MS Analysis

A total of 100 mg of extract was mixed with 80% methanol for 90 s and subjected to ultrasonication at 4 °C for 30 min. Sonicated samples were kept at −20 °C for 1 h, subjected to vortex for 30 s, and incubated at 4 °C for 30 min. All the samples were subjected to centrifugation for 15 min at 12,000 rpm and 4 °C. From this, 200 μL of the supernatant was mixed with 5 μL of DL-o-chlorophenylalanine (0.14 mg/mL) and transferred to a vial for LC-MS analysis.

### 3.6. UPLC–ToF–MS Analysis Conditions

Separation of the components was performed by Acquity UPLC (Waters, Milford, MA, USA) coupled with Q Exactive MS (Thermo Scientific, Mississauga, ON, Canada) and screened with ESI-MS. The LC system consisted of an Acquity HSS T3 UPLC column (100 mm × 2.1 mm × 1.8 μm). The mobile phases used for analysis include 0.05% formic acid in water as solvent A and acetonitrile as solvent B. The gradient elution program used for separation was (0–1 min, 95% A; 1–12 min, 95–5% A; 12–13.5 min, 5% A; 13.5–13.6 min, 5–95% A; 13.6–16 min, 95% A) with a flow rate of 0.3 mL/min. The column was constantly maintained at 40 °C, and the sample manager was set at 4 °C. Mass spectrometry parameters used for separation were ESI+, heater temp 300 °C; sheath gas flow rate, 45 arb; aux gas flow rate, 15 arb; sweep gas flow rate, 1 arb; spray voltage, 3.0 kv; capillary temp, 350 °C; s-lens rf level, 30%; ESI–: heater temp 300 °C, sheath gas flow rate, 45 arb; aux gas flow rate, 15 arb; sweep gas flow rate, 1 arb; spray voltage, 3.2 kv; capillary temp, 350 °C; s-lens rf level, 60%.

### 3.7. Data Processing

The raw data obtained from the analysis was pre-processed for baseline correction, deionization, smoothing, splitting, and deconvolution using Progenesis QI software (Waters, Mississauga, ON, Canada). A data matrix composed of retention time, sample description (triplicates of each cherry cultivar), *m*/*z* ratio, and abundance was analyzed in MetaboAnalyst 5.0 software.

Further, the raw data were acquired and aligned using Compound Discover (3.0, Thermo Fisher, Mississauga, ON, Canada) based on the *m*/*z* value and retention times for the ion signals. For multivariate analyses, ions from ESI− and ESI+ were combined and imported into the SIMCA-P program (version 14.1). Multivariate and statistical analyses, such as partial least-square–discriminant analysis (PLS-DA), principal component analysis (PCA), and heatmap generation were performed with MetaboAnalyst 5.0 software. This approach combines unsupervised (PCA) and supervised (PLS-DA) methods for the visualization, outlier detection, and identification of potential biomarkers in high-dimensional data. The final biomarkers are filtered and confirmed using a combination of VIP values (VIP > 1) and statistical tests such as *t*-test (*p* < 0.05).

### 3.8. Metabolite Identification

After refining the MS spectra data, the most significant metabolites contributing to different bioactive components were identified by mass ratio, mass spectra, metabolomics databases, and literature. The identified metabolites were searched in the human metabolome database (HMDB) 2022, tandem mass spectra of metabolites using Metfrag, and Massbank. No standards were used for the identification of components. The pathway map was deciphered using well-annotated HMDB compounds with pathway libraries and the KEGG database.

### 3.9. Statistical Analysis

The bioactive component analysis is the result of three different extracts. Statistical comparison of all the parameters was performed using pairwise comparison using Tukey’s HSD test at (*p* ≤ 0.05). Statistical analysis was performed by one-way or multi-way ANOVA using R software (v.1.2.1.) with agricolae package. All the figures were generated using the statistical software Graphpad prism (Prism 5.01 Inc., La Jolla, CA, USA).

## 4. Conclusions

Sour cherry has great potential for use as a functional food due to its high antioxidant activity. Untargeted metabolomics identified a total of 473 metabolites of different functional chemical groups like organic acids, phenolics, flavonoids, anthocyanins, amino acids, and sugars. Among the four different cultivars tested, the dark-flesh cultivars had a higher level of bioactive metabolites than the light-flesh cultivars. Further, the metabolic profiling revealed that the chemical composition of sour cherry is predominated by fatty acids, organic acids, benzenoids, polyketides, and organo-heterocyclic compounds. Seven sour cherry metabolites (allysine, biochanin A, hydroxybutyric acid, acetylacrylic acid, 2,4-hexadienedial, indole, and chlorogenic) were identified to influence the antioxidant activity. The relative content of these metabolites was found to be significantly higher in dark-flesh cultivars. The results obtained in this study provide insight into the food, pharmaceutical, and nutraceutical industries to develop sour cherry as a functional food. This study will also provide help to plant breeders to develop new genotypes and cultivars with enhanced nutritional potential.

## Figures and Tables

**Figure 1 plants-13-01511-f001:**
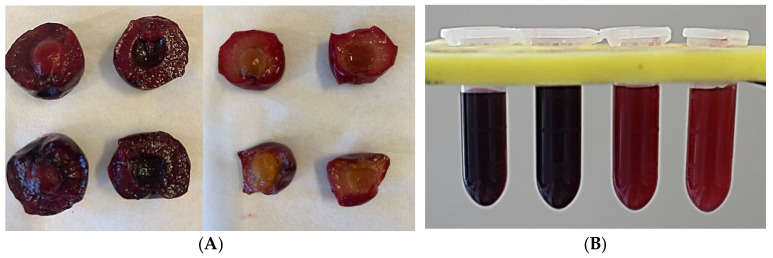
(**A**) Fruits of dark-flesh cultivars (‘Heimann R’ and ‘Gorsemska’—top left and bottom left) and light-flesh cultivars (‘Montmorency’ and V70142—top middle and bottom middle). (**B**) Freshly extracted juice of ‘Heimann R’, ‘Gorsemska’, ‘Montmorency’, and V70142—from left to right.

**Figure 2 plants-13-01511-f002:**
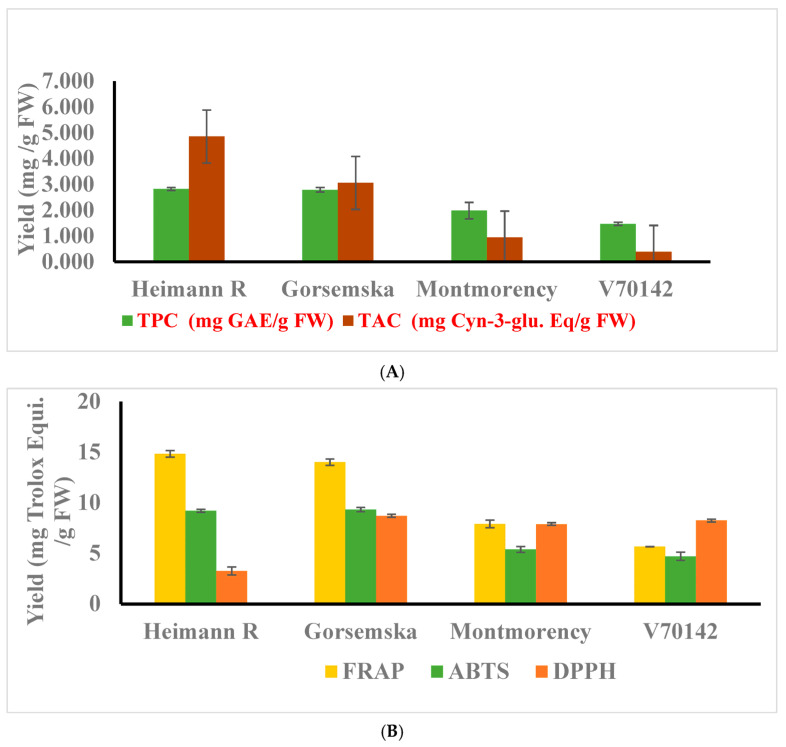
(**A**) Total phenolic, and anthocyanin contents were determined. (**B**) Antioxidant activities of cherry cultivars were determined using FRAP, ABTS, and DPPH assays. The experiments were carried out in three biological replicates, and each replicate was repeated three times (*n* = 9). Data represent the mean values ± SD (*n* = 3).

**Figure 3 plants-13-01511-f003:**
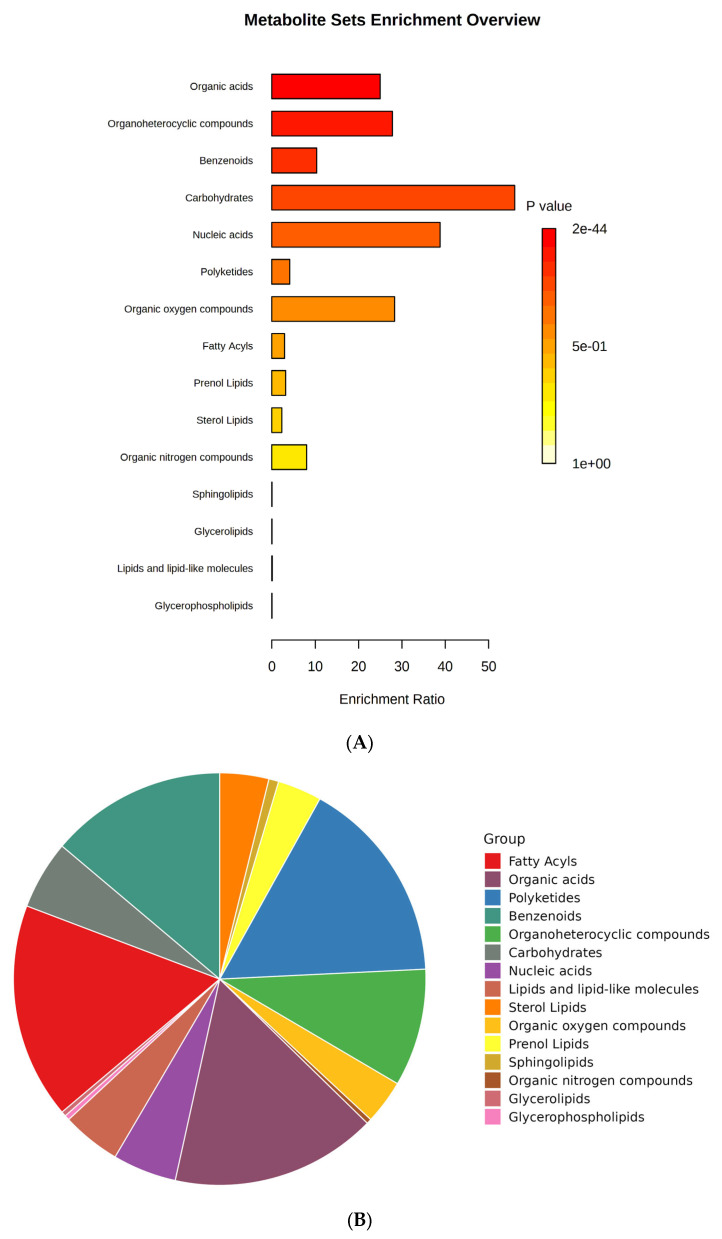
(**A**) Bar chart and (**B**) pie chart of the chemical classification of cherry metabolites using metabolite set enrichment analysis (MSEA). Colors in the bar plot describe the *p*-value. The red and orange colors signify the high and low values, respectively. The lines indicate the enrichment ratio, which was computed by hits/expected, where hits = observed hits and expected = expected hits. The colors in the interactive pie chart designate each chemical group relative to the total number of compounds.

**Figure 4 plants-13-01511-f004:**
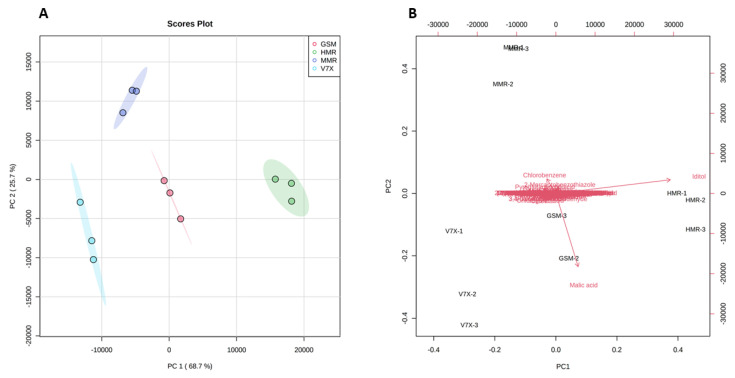
Principal components analysis (PCA) 2D score plot (**A**) and biplot (**B**) of the cherry metabolites. The different short abbreviations in the biplot manifest the scores of the observations (i.e., cherry cultivars). The vectors that point toward the same direction correspond to the variables (i.e., metabolites) with similar response profiles. The red represents GSM_ ‘Gorsemska’; green represents HMR_ ‘Heimann R’; purple represents MMR_ ‘Montmorency’; blue represents V7X_ V70142.

**Figure 5 plants-13-01511-f005:**
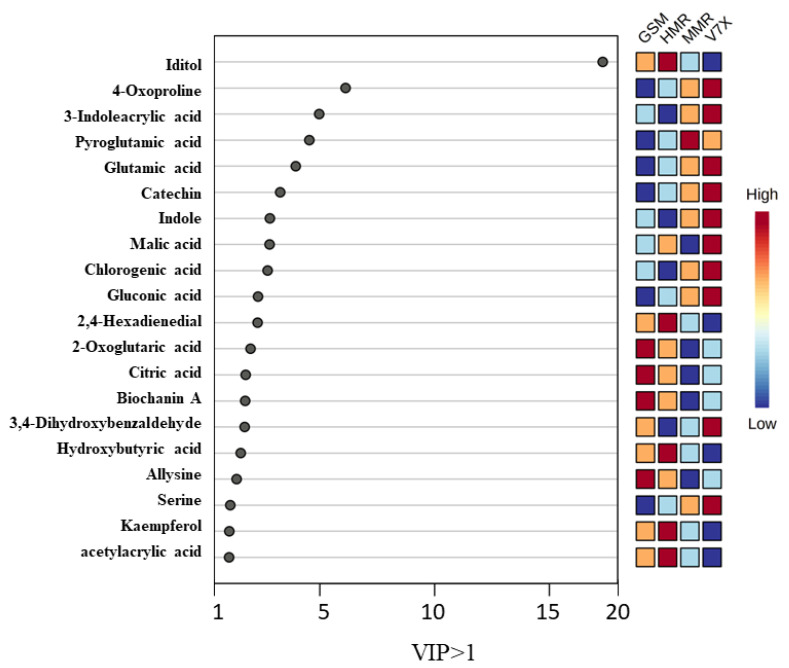
The weighted sum of absolute regression coefficients of candidate metabolites (VIP > 1.0) of cherry cultivars. The colored boxes on the right indicate the relative concentrations of the corresponding metabolite in each group under study.

**Figure 6 plants-13-01511-f006:**
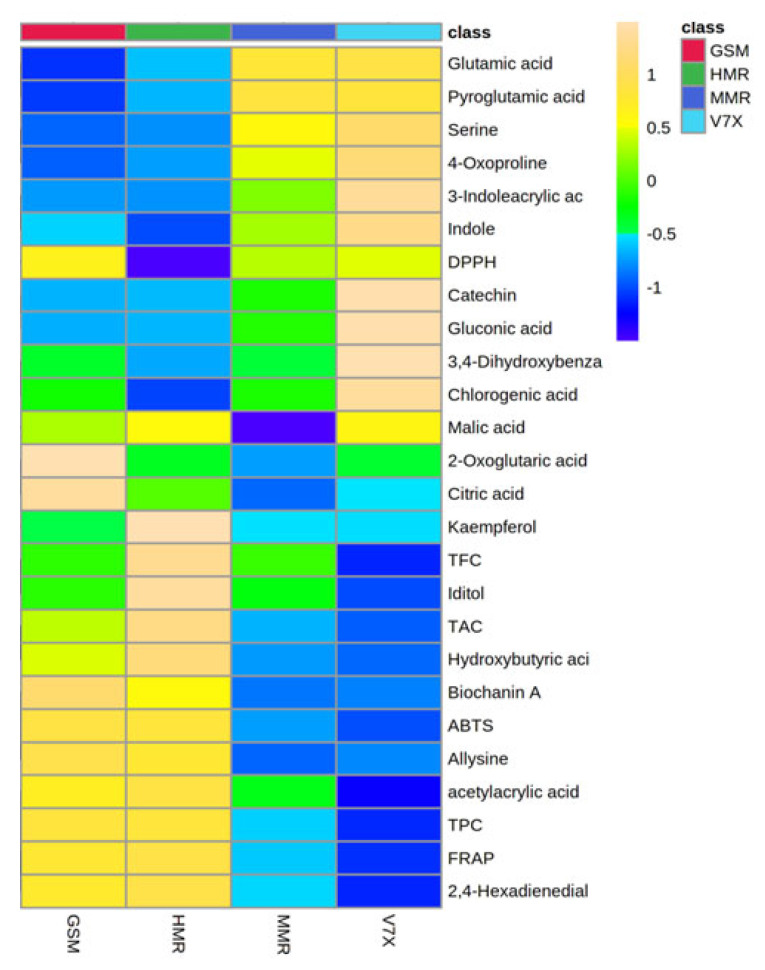
Heatmap analysis of candidate metabolites (VIP > 1.0) obtained by partial least-squares–discriminant analysis (PLS-DA) of various metabolites of cherry cultivars. Each column refers to the cherry cultivars, and each row indicates the metabolites, TPC, TAC, and antioxidant activities. The red and blue colors in the plot describe high and low intensities, and the values range from −1 to +1. The greater the red color intensity (from +1 to +2 values), the higher the metabolite contents and antioxidant activities; in contrast, the greater blue color intensity (from −1 to −2 values) represents lower metabolite contents and antioxidant activities. GSM: ‘Gorsemska’; HMR: ‘Heimann R’; MMR: ‘Montmorency’; V7X: V70142.

**Figure 7 plants-13-01511-f007:**
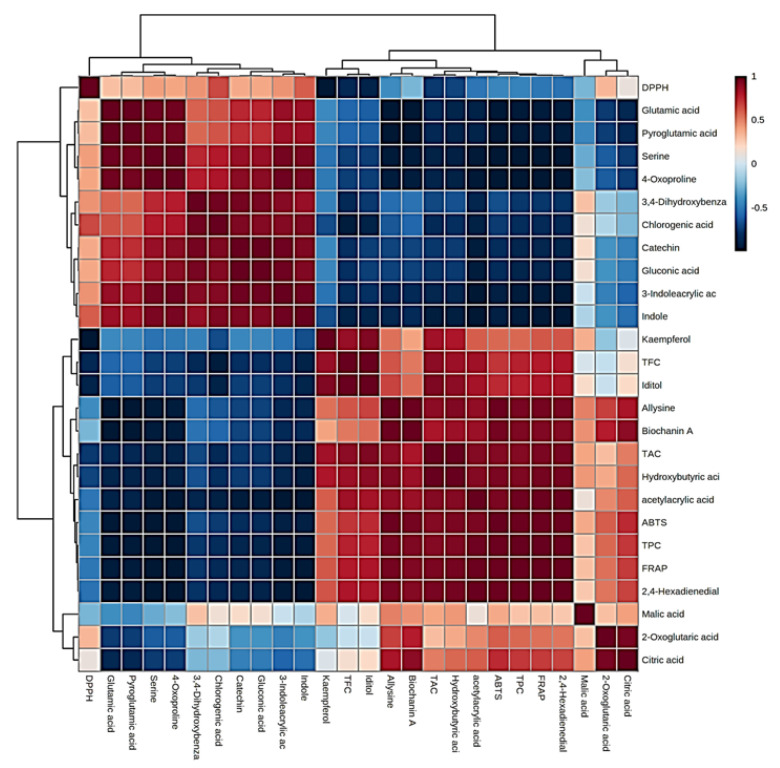
Heatmap of Pearson’s correlation between candidate metabolites (VIP > 1.0) with antioxidant activities of cherry cultivars. Correlation values range from −1 to +1. The values close to +1 represent the higher positive correlation, whereas values closer to zero mean there is no linear trend between the variables; values close to −1 represent the negative correlation between variables.

**Table 1 plants-13-01511-t001:** TPC, TAC, and antioxidant activity (FRAP, ABTS, and DPPH) of genotypes used in this study.

Genotypes	TPC (mg GAE/g FW) *	Total Anthocyanin (mg Cyn-3-glu. Eq/g FW) **	FRAP (mg Trolox/g FW)	ABTS (mg Trolox/g FW)	DPPH (mg Trolox/g FW)
‘Heimann R’	2.821 ± 0.062	4.857 ± 0.154	14.834 ± 0.320	9.192 ± 0.149	3.242 ± 0.387
‘Gorsemska’	2.789 ± 0.088	3.055 ± 0.064	14.008 ± 0.316	9.319 ± 0.213	8.693 ± 0.152
‘Montmorency’	1.982 ± 0.318	0.939 ± 0.063	7.900 ± 0.379	5.368 ± 0.296	7.880 ± 0.130
V70142	1.466 ± 0.064	0.383 ± 0.064	5.648 ± 0.018	4.690 ± 0.404	8.232 ± 0.142

* GAE—gallic acid equivalent; ** Cyn-3-glu Eq/g FW—cyanidin-3-glucoside equivalent/gram fresh weight.

## Data Availability

Data generated or analyzed during this study are provided in full within the published article.

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
