# Peer review of "Comparative Antioxidant Activity and Untargeted Metabolomic Analyses of Sour Cherry Cultivars Based on Ultra-Performance–Time of Flight–Mass Spectrometry"

_plants, 2024, doi:10.3390/plants13111511_

Round 1

Reviewer 1 Report

Comments and Suggestions for Authors

1. Introduction part: 

- Please, rewrite the introduction part, to contain ongoing information about the presence of phenolic substances in sour cherries, and then to describe their positive effects.

2. Results and discussion:

- It is necessary to redo graphs due to unreadability. Too small letters are used

- In the Figure 1B, 1C, is important to include statistically significant differences into the graphs.

- Figure 1, n=3 or n=9? Please explain.

3. Material and Methods:

- Please include information about the location of cherry trees.

- For how long was the fruit stored at -80°C?

- I suggest to describe the estimation of TPC in the manuscript.

- What are the minor modification that were made in TFC measurement? Please describe.

4. I recommend adding subsection statistics with description. 

5. There are some typomistakes in the manuscript.

6. Please check information about authors, and double check formating.

Author Response

The manuscript seems to be written in rush. Many gaps in the content are seen as the story does not flow consistently. Authors often use vernacular expressions, and sentences sometimes rest understated. Some scientific expressions are inconsistently used and sometimes a reader may have an impression even the authors themselves didn't understand what they wanted to say. Many syntax and typo errors are noted within the manuscript. Some places of possible intervention are highlighted in the attached file.

Response:

The authors would like to thank the reviewer for the comments. Introduction, Results and Discussions and Material and methods have been modified. All the highlighted areas have been addressed and highlighted in red color.

2) Abstract is quite uninformative. It is vague why the authors have done the study anyway. More information on the results obtained and novelty of the study should stand in Abstract.

Response:

The authors would like to thank the reviewer for the comments. Abstractt has been modified.

3) The authors claim this is the first detailed report of the metabolic profiling of sour cherry, but this is not true (the details are provided in the attached file). Moreover, it seems that the authors are poorly familiar with the literature published on the same topic. Some recent articles such as 10.1016/j.jfca.2015.10.002 and 10.1021/jf980936f weren't even listed in the reference list.

Response:

The authors would like to explain here that this is the first detailed study of sour cherry cultivars grown in Canada as well as North America.  The authors have mentioned in the manuscript and highlighted in red in Line 62.

- The authors talk about dark-color flesh and light color flesh cultivars but it seems they mean exocarp, since they presented source cherry fruit in Fig. 1A in full, without cross-section, but concluded the flesh color upon the exocarp color, which may not be the same.

Response:

The authors would like to explain here that sour cherry whole fruit is taken in study without the cross-section. But the color of the extract and internal material was also based on the flesh color and defined the cultivars as dark and light-colored cultivars.

- The authors introduced the term "biomarker metabolites" in chapter 2.3, but never described what they meant under this expression.

Response:

The authors have explained the term in section 3.4.1 and highlighted in red.

- For other specific comments, please see the attached file. peer-review-35695087.v1.pdf

Response:

The authors have modified the whole manuscript and highlighted in red.

Comments on the Quality of English Language

The manuscript seems to be written in rush. Many gaps in the content are seen as the story does not flow consistently. Authors often use vernacular expressions, and sentences sometimes rest understated. Some scientific expressions are inconsistently used and sometimes a reader may have an impression even the authors themselves didn't understand what they wanted to say. Many syntax and typo errors are noted within the manuscript. Some places of possible intervention are highlighted in the attached file.

Response:

The authors have modified the whole manuscript and highlighted in red.

Reviewer 2 Report

Comments and Suggestions for Authors

The manuscript submitted to Plants titled "Comparative antioxidant activity and untargeted metabolomic 2 analyses of sour-cherry cultivars based on UPLC-TOF-MS" contains valuable information on inter-cultivar variation of specialized metabolites produced by sour cherry. However, although authors report 473 metabolites identified by UPLC-ESI-ToF/MS, a reader was never introduced with those compounds and this is the main reason for suggesting rejection. Other major reasons are as follows:

- The manuscript seems to be written in rush. Many gaps in the content are seen as the story does not flow consistently. Authors often use vernacular expressions, and sentences sometimes rest understated. Some scientific expressions are inconsistently used and sometimes a reader may have an impression even the authors themselves didn't understand what they wanted to say. Many syntax and typo errors are noted within the manuscript. Some places of possible intervention are highlighted in the attached file.

- Abstract is quite uninformative. It is vague why the authors have done the study anyway. More information on the results obtained and novelty of the study should stand in Abstract.

- The authors claim this is the first detailed report of the metabolic profiling of sour cherry, but this is not true (the details are provided in the attached file). Moreover, it seems that the authors are poorly familiar with the literature published on the same topic. Some recent articles such as 10.1016/j.jfca.2015.10.002 and 10.1021/jf980936f weren't even listed in the reference list.

- The authors talk about dark-color flesh and light color flesh cultivars but it seems they mean exocarp, since they presented source cherry fruit in Fig. 1A in full, without cross-section, but concluded the flesh color upon the exocarp color, which may not be the same.

- There's something substantially wrong with the TPC and TFC content in Fig 1B. Being phenolics themselves, the content of flavonoids can only be a subset of phenolics. From what I see from the figure, TFC exceeds TPC for up to 30 times, which cannot be possible if assays were done properly.

- The authors introduced the term "biomarker metabolites" in chapter 2.3, but never described what they meant under this expression.

- For other specific comments, please see the attached file.

Comments on the Quality of English Language

The manuscript seems to be written in rush. Many gaps in the content are seen as the story does not flow consistently. Authors often use vernacular expressions, and sentences sometimes rest understated. Some scientific expressions are inconsistently used and sometimes a reader may have an impression even the authors themselves didn't understand what they wanted to say. Many syntax and typo errors are noted within the manuscript. Some places of possible intervention are highlighted in the attached file.

Author Response

Reviewer 2:

General Answers:

The manuscript submitted to Plants titled "Comparative antioxidant activity and untargeted metabolomic 2 analyses of sour-cherry cultivars based on UPLC-TOF-MS" contains valuable information on inter-cultivar variation of specialized metabolites produced by sour cherry. However, although authors report 473 metabolites identified by UPLC-ESI-ToF/MS, a reader was never introduced with those compounds, and this is the main reason for suggesting rejection. Other major reasons are as follows:

With all due respect to the reviewer, I don’t think that will constitute a reason to REJECT a manuscript. All it requires is to ask the authors to add it. Since the table of 473 compounds will be way too large for a manuscript, it is added as a supplementary file.

- The manuscript seems to be written in rush. Many gaps in the content are seen as the story does not flow consistently. Authors often use vernacular expressions, and sentences sometimes rest understated. Some scientific expressions are inconsistently used and sometimes a reader may have an impression even the authors themselves didn't understand what they wanted to say. Many syntax and typo errors are noted within the manuscript. Some places of possible intervention are highlighted in the attached file.

As detailed below, we have corrected the passages/sentences where the reviewer felt it needs intervention.

- Abstract is quite uninformative. It is vague why the authors have done the study anyway. More information on the results obtained and novelty of the study should stand in Abstract.

The abstract has been modified to reflect the concerns.

- The authors claim this is the first detailed report of the metabolic profiling of sour cherry, but this is not true (the details are provided in the attached file). Moreover, it seems that the authors are poorly familiar with the literature published on the same topic. Some recent articles such as 10.1016/j.jfca.2015.10.002 and 10.1021/jf980936f weren't even listed in the reference list.

We claim it is a ‘detailed report’. Yes, there are other reports but not a ‘detailed report’ on the entire metabolome, as we claim. Yet, we have removed the first report claim.

- The authors talk about dark-color flesh and light color flesh cultivars, but it seems they mean exocarp, since they presented source cherry fruit in Fig. 1A in full, without cross-section, but concluded the flesh color upon the exocarp color, which may not be the same.

I think the reviewer is new to cherry. Cherry exocarp remains more or less same red colour barring those white cherries where they lack anthocyanins partially or fully. The flesh color changes to light (yellowish) or dark (wine red) and that is reflected on the phenotypic look. We didn’t conclude as the reviewer suggested based on the exocarp but we have seen the flesh as well and those two varieties that are used as dark fleshed variety are in the literature for a long time and is commercially grown in parts of Europe. But the fact is these varieties are in practice for a long time and known for their dark flesh.

- There's something substantially wrong with the TPC and TFC content in Fig 1B. Being phenolics themselves, the content of flavonoids can only be a subset of phenolics. From what I see from the figure, TFC exceeds TPC for up to 30 times, which cannot be possible if assays were done properly.

While we fully agree that Flavonoids are part of total phenolics and thus in reality they cannot be more than TPC, what we are providing here is estimates (not actual values, which one can never be certain of) and there is general consensus among researchers that occasionally you see such differences depending on the standard used. Had we used rutin/catechin instead of quercetin we might have seen lower TFC than TPC. Nevertheless, that is not a major focus of this study and hence we have removed that data (Please see revised Fig 1B) and any related references in the text.

- The authors introduced the term "biomarker metabolites" in chapter 2.3, but never described what they meant under this expression.

The overarching term ‘biomarker’ has been removed now.

Answers to specific comments by the reviewer 2

  1. Introduction part: 

Please, rewrite the introduction part, to contain ongoing information about the presence of phenolic substances in sour cherries, and then to describe their positive effects.

Response:

The authors would like to thank the reviewer for the comments. Authors have modified the introduction section and highlighted the changes in red color.

  1. Results and discussion:

- It is necessary to redo graphs due to unreadability. Too small letters are used.

Response:

The graphs have been modified and added again with improved visibility.

- In the Figure 1B, 1C, is important to include statistically significant differences into the graphs.

Response:

The authors have added the significant differences in the graphs.

- Figure 1, n=3 or n=9? Please explain.

Response:

The authors would like to thank the reviewer for the comments. n=3 as it is the mean of 3 samples

  1. Material and Methods:

- Please include information about the location of cherry trees.

Response: The authors have included the location information in Cherry samples in Line 270-271 and highlighted with red color.

- For how long was the fruit stored at -80°C?

Response: The authors have included the storage time in line 275.

- I suggest describing the estimation of TPC in the manuscript.

Response: The authors have described the TPC estimation method and highlighted in red from Line 292-304.

- What are the minor modification that were made in TFC measurement? Please describe.

Response: The authors have described the TFC estimation method and highlighted in red from Line 292-304.

  1. I recommend adding subsection statistics with description. 

Response: The authors have added the statistical analysis in section 3.4.5.

  1. There are some typo mistakes in the manuscript.

Response: The authors have thoroughly read the manuscript and corrected the typos.

  1. Please check information about authors, and double check formatting.

Response:

The authors have checked the authors information and formatting.

Reviewer 3 Report

Comments and Suggestions for Authors

Comments on the article paper (nutrients) entitled "Comparative antioxidant activity and untargeted metabolomics analyses ot sour-cherry cultivar based on UPLC-TOF-MS.” by Kaur et al.

The paper studies the correlation between antioxidant properties and the main identified metabolites in four Sour-cherry cultivars.

Despite the its important statistical work and technical complexity, the paper is easy to read and understand and its structure and distribution is appropriate to publish in this journal, at the same time the subject matter also is appropriate. The English language is satisfactory, the manuscript is well presented and the information and data showed are very interesting and also represent an important advance in the knowledge and understanding of the relationship between the structure of metabolites present in food and their biological activity.

I consider that this manuscript could be accepted for publication in its present form.

Author Response

Comments on the article paper (nutrients) entitled "Comparative antioxidant activity and untargeted metabolomics analyses ot sour-cherry cultivar based on UPLC-TOF-MS.” by Kaur et al.

The paper studies the correlation between antioxidant properties and the main identified metabolites in four Sour-cherry cultivars. Despite the its important statistical work and technical complexity, the paper is easy to read and understand and its structure and distribution is appropriate to publish in this journal, at the same time the subject matter also is appropriate. The English language is satisfactory, the manuscript is well presented and the information and data showed are very interesting and also represent an important advance in the knowledge and understanding of the relationship between the structure of metabolites present in food and their biological activity.

I consider that this manuscript could be accepted for publication in its present form.

Response:

The authors would like to thank the reviewer for the comments.

Round 2

Reviewer 2 Report

Comments and Suggestions for Authors

I regret to see a slightly adverse tone in authors' response. Please note that I spent many hours of my spare time and didn't spare my professional knowledge to render advises to the authors how to further improve their manuscript, which will further possibly result in more citations and more serious comprehension of their (not my) work.

Regarding the authors' answer to the reviewer's comment on the table with 473 compounds: with all due respect to the authors, this was not a (one) reason for rejection but was stated as the MAIN one. Other reasons were stated below and the authors might conclude while reading the report that the suggestion REJECT was not inferred solely from the missing table. Moreover, the authors might ask themselves why a reviewer should ask for data. Is this his/her responsibility? Isn't it authors' responsibility to provide all supporting data during submission? Otherwise, one may conclude the results were fabricated. Finally, I cannot access the required table in the revised manuscript. Are the authors sure they have submitted it this time? Lines 105 and 106 do not reference to such a table, so I must conclude the authors never intended to submit it.

To the reviewer's comment "Abstract is quite uninformative. It is vague why the authors have done the study anyway. More information on the results obtained and novelty of the study should stand in Abstract.", the authors responded with "The abstract has been modified to reflect the concerns." which do not hold the true, as it has been left as it was.

Regarding the authors' reply on the "detailed report". It is quite a subjective assessment what a "detailed report" should represent. I'm sure that the authors of the two suggested references would claim their reports were quite detailed. Moreover, "entire metabolome" cannot be assessed by a single analytical method, as used in this study.

Regarding the authors' statement that "the reviewer is new to cherry" it may be true. However, then it should be the authors of the study "Genetic diversity in a core collection of Iranian sour cherry" (DOI: 10.1590/1519-6984.273386) who reported yellow flesh but red skin (exocarp) of the populations G102 and G453 (they are not "white cherries"). The problem is that the authors presented only the outer look of the fruits but not cross-section (they might learn from the cited paper how to do it). Finally, a reader or a reviewer should not guess which flesh color the authors "have seen" but which color is presented in the figure that is visible to readers. Simple shifting the whole Figure 1 into the M&M section cannot be an acceptable solution.

Cultivar names were not put under the single quotation marks, as suggested in the previous review round, although the authors claimed they did it.

To the reviewer's comment to additionally explain the term "biomarker metabolites", the authors answered with "The overarching term ‘biomarker’ has been removed now." which does not hold the true as it stands in the subtitle in L160.

Regarding TPC and TFC values, the authors claimed "that is not a major focus of this study and hence we have removed that data"; the same data were not removed but shifted to M&M as Figure 6B.

Author Response

We thank the reviewer for the constructive comments and we have addresses the comments in the attached report.

Round 3

Reviewer 2 Report

Comments and Suggestions for Authors

The authors need to thoroughly and meticulously check the whole text as being written in a rather easygoing way. For example, the main analytical methodology is termed UPLC-QToF/MS, UPLC-qToF -MS, and UPLC-TOF-MS across the text, which is unacceptable. Cultivar names haven't still been put under single quotation marks throughout the text. Typographical errors are quite frequent, e.g., fatty acyls (?!) in Figure 3. Random word capitalization is also frequent. I strongly advise having the manuscript proofread by a native English speaker or a professional editing agency. Major problem: Figure 2A reportedly present "Total phenolic, flavonoid, and anthocyanin contents", but I see just TPC and TAC.

Comments on the Quality of English Language

Typographical errors and random word capitalization are quite frequent.

Author Response

Here are the answers to the comments posed by the reviewer:

The authors need to thoroughly and meticulously check the whole text as being written in a rather easygoing way.

I have checked the manuscript completely today and have made a few very minor corrections, which in the normal sense would not have made a huge difference in the total context of the paper.

For example, the main analytical methodology is termed UPLC-QToF/MS, UPLC-qToF -MS, and UPLC-TOF-MS across the text, which is unacceptable.

Understood and a valid minor comment, that would be caught during typesetting for sure, even if we miss here. We have corrected it all to UPLC-TOF-MS, although TOF should ToF, but we are doing it at the insistence of this reviewer. Some of the changes had happened during the many revisions and hence slipped my attention as I always insist on consistency to my students.

Cultivar names haven't still been put under single quotation marks throughout the text.

Checked it thoroughly using 'Find and Replace' function even- corrected all. Again, this would be corrected in the typeset before publication in normal circumstances.

Typographical errors are quite frequent, e.g., fatty acyls (?!) in Figure 3.

It is actually Fatty Acyl - The reviewer is thinking we made a typo as acyl and not acid. Fatty acyl group includes fatty acids. It is not a typo and other than that I did not find any typo as mentioned.

Random word capitalization is also frequent. I strongly advise having the manuscript proofread by a native English speaker or a professional editing agency.

Again -other than places where capitalization is a must there is no unwanted capitalization to my knowledge. Can someone define me what is a Native English speaker please? FYI – I have been offered by some ‘Professional Editing Agency’ in the past as well -to do some editing for pay, which I declined. I have my strong reservations on that particular comment about English – but do not want to write it.

Major problem: Figure 2A reportedly present "Total phenolic, flavonoid, and anthocyanin contents", but I see just TPC and TAC.

I think in the past 2 revisions (at least) we have agreed upon and removed anything related to TFC based on the comments from the academic editor. Fig 2A is thus presenting only TPC and TAC. I did not see anything in the resuts related to TFC in the last version submitted and I have removed the inadvertent addition in the methods on the reference to TFC, as indicated by the editor in the last revision.

Thank you

Jay